# Glucose-coated superparamagnetic iron oxide nanoparticles prepared by metal vapor synthesis can target GLUT1 overexpressing tumors: In vitro tests and in vivo preliminary assessment

**Daniele Barbaro**[1☯]*, **Lorenzo Di Bari**[2☯], **Valentina Gandin**[3☯], **Cristina Marzano**[3], **Andrea Ciaramella**[4], **Michele Malventi**[4], **Claudio Evangelisti**[5]

1 U.O. Endocrinology, General Hospital, Livorno, Livorno, Italy, 2 Department of Chemistry and Industrial Chemistry, University of Pisa, Pisa, Italy, 3 Department of Pharmaceutical and Pharmacological Sciences, University of Padova, Padova, Italy, 4 U.O. Radiology General Hospital Livorno, Livorno, Italy, 5 Institute for the Chemistry of OrganoMetallic Compounds, Italian National Council for Research (ICCOM-CNR), Pisa, Italy

☯ These authors contributed equally to this work.
* danielebarbaro@katamail.com

**Data Availability Statement:** All relevant data are within the paper.

## Abstract

Superparamagnetic iron oxide nanoparticles (SPIONs) coated with glucose (Glc-SPIONs) were prepared by a new approach called Metal Vapor Synthesis (MVS) and their morphological/structural features were investigated by transmission electron microscopy (TEM) and dynamic light scattering. TEM analysis revealed the presence of small roundish crystalline iron oxide nanoparticles in the organic amorphous phase of glucose, The particles were distributed in a narrow range (1.5 nm—3.5 nm) with a mean diameter of 2.7 nm. The hydrodynamic mean diameter of the Glc-SPIONs, was 15.5 nm. From 4 mg/mL onwards, there was a constant level of positive contrast in a T1-weighted sequence. *In vitro* experiments were performed in three cell lines: pancreatic cancer (PSN-1), human thyroid cancer (BCPAP), and human embryonic kidney non-tumor cells. We evaluated GLUT1 expression in each cell line and demonstrated that the exposure time and concentration of the Glc-SPIONs we used did not affect cell viability. PSN-1 cells were the most effective at internalizing Glc-SPIONs. Although significantly higher than the control cells, a lower Fe content was detected BCPAP cells treated with Glc-SPIONs. To confirm the involvement of GLUT1 in Glc-SPIONs internalization, cellular uptake experiments were also conducted by pre-treating cancer cells with specific GLUT1 inhibitors, All the inhibitors reduced the cancer cell uptake of Glc-SPIONs *In vivo* tests were performed on mice inoculated with Lewis lung carcinoma. Mice were treated with a single i.v. injection of Glc-SPION and our results showed a great bioavailability to the malignant tissue by the i.v. administration of Glc-SPIONs. Glc-SPIONs were efficiently eliminated by the kidney. To the best of our knowledge, our study demonstrates for the first time that Glc-SPIONs prepared with MVS can be electively internalized by tumor cells both *in vitro* and *in vivo* by exploiting one of the most universal metabolic anomalies of cancer.

**Funding:** The authors received no specific funding for this work.

**Competing interests:** The authors have declared that no competing interests exists.

## Introduction

A rapidly developing area of oncology is cancer targeting by means of nano-based delivery for organic or inorganic molecules and systems, such as drugs or metals. The delivery carriers can be organic polymers, as well as small organic molecules, which endow the complex with novel properties, thus decreasing its toxicity, and improving its pharmacokinetic features [1–7]. Very recently, self-assembly has enabled peptides to be manipulated in order to obtain self-assembling systems, adorned with functional features targeting cancer cells, thus, leading to the specific recognition of neoplasia [8].

As an alternative to more sophisticated methods, our study investigated the possibility of functionalizing metal nanoparticles (MNs) with a simple organic molecule, such as glucose, which can in theory be a universal ligand for cancer cells.

MNs have already found multiple uses in biomedicine, both for diagnosis and therapy [9,10].

The most studied and developed application of MNs is as contrast agents in magnetic resonance imaging (MRI). For this purpose, the metal (oxide) core of the MNs has to be coated with organic or polymeric ligands. This organic shell prevents the release of iron or any other potentially toxic or noxious metal and ensures biocompatibility and a better stability to the colloid [11]. In fact these polymeric coatings create a barrier between the MN core and the biological fluid, resulting in extracellular fluid contrast agents, an application, where gadolinium (Gd) complexes are most widely used [12,13].

One of the most appealing features of MNs is the possibility of functionalizing them with specific ligands in order to obtain organ- or tissue-specific agents [10,14–16]. Notably, the endocytosis of abiotic nanomaterials such as MNs inside cancer cells represents the basis for specific cancer detection and above all cancer therapy [17–20]. In fact, among MNs, superparamagnetic nanoparticles, and especially superparamagnetic iron oxide nanoparticles (SPION), can potentially be used, by exploiting their ability to be heated under alternating magnetic fields [21–23]. In addition, TAT peptide-coated iron oxide nanoparticles have been shown to enhance radiation therapy by generating reactive oxygen species *in vitro* [24].

For hyperthermic therapy, polymer-based SPIONs are used, but unfortunately they have to be targeted at the tumor tissue by direct injection [25,26]. Selectively delivering functionalized SPIONs into tumor tissue by systemic administration represents a significant step forward. SPIONs thus need to be functionalized with ligands that can recognize neoplastic cells and promote the internalization of the SPIONs themselves.

Several ligands for MNs have been reported including some specific ligands for specific cancers [19,27,28]. It is well known that one of the basic characteristics of tumor cells is that they are greedy for glucose. This fundamental aspect, known as the Warburg effect, is already the basis of one of the most important tools in oncology: 18-fluoro deoxiglucose positron emission tomography (18 FDG PET), which is usually associated with computerized tomography (18 FDG PET/CT) [29].

Building glucose-coated SPIONs coating is key in targeting neoplastic cells via a systemic route, although this is only possible with ultra-small SPIONs. This size requirement is instrumental to their internalization within the tumor cells, thanks to the active role of membrane transporters or receptors.

We recently presented small glucose-coated iron oxide SPIONs (Glc-SPIONs), prepared through an innovative technique, called Metal Vapor Synthesis (MVS) [33]. These Glc-SPIONs are homogeneous, with a mean diameter of 2.7 nm, and surrounded by a thick layer of glucose, reaching an apparent hydrodynamic diameter of 13 nm [30].

In our previous study, the Glc-SPIONs were electively internalized in a pancreatic adeno-carcinoma cell line (BxPC3) [31].

After our paper, in another work by a different group, Glc-SPIONs were shown to be internalized by GLUT1, in a line of mammary breast cancer [32]. Some studies have also reported that 2-deoxy-glucose capped gold nanoparticles can be internalized by different cancer cell lines [33], although, to date, no *in vivo* experiment has been reported.

To the best of our knowledge, this paper reports the results of tests conducted both in vitro and in vivo for the first time. In vitro, we confirmed the specific uptake of our Glc-SPIONs in several cancer lines with different over-expressions of GLUT1 transporter, using different GLUT1 inhibitors. In vivo, we demonstrated the possibility that, after systemic administration, our Glc-SPIONs were delivered and electively internalized by the tumor.

## Materials and methods

### Glucose-coated superparamagnetic iron oxide nanoparticles (Glc-SPIONs)

The glucose-coated iron oxide nanoparticles through metal vapor synthesis (MVS) were prepared by Advanced Catalysts S.R.L. (Livorno, Italy), following the procedure reported above [33].

The preparation was conducted in a high vacuum (about $10^{-5}$ mBar), by placing approximately 300 mg of Fe in an alumina-coated tungsten crucible heated by the Joule effect, using a generator with a maximum power of 2 kW. The outer walls of the reactor were cooled through a liquid nitrogen bath (Fig 1).

During Fe sublimation, 100 mL acetone (Aldrich, $\geq$99.5%) was co-vaporized, leading to the co-condensation of Fe atoms and acetone (solvated metal atoms, SMA) on the cold vessel walls. After terminating the heating, the vessel was removed from the $N_2(l)$ bath, and the

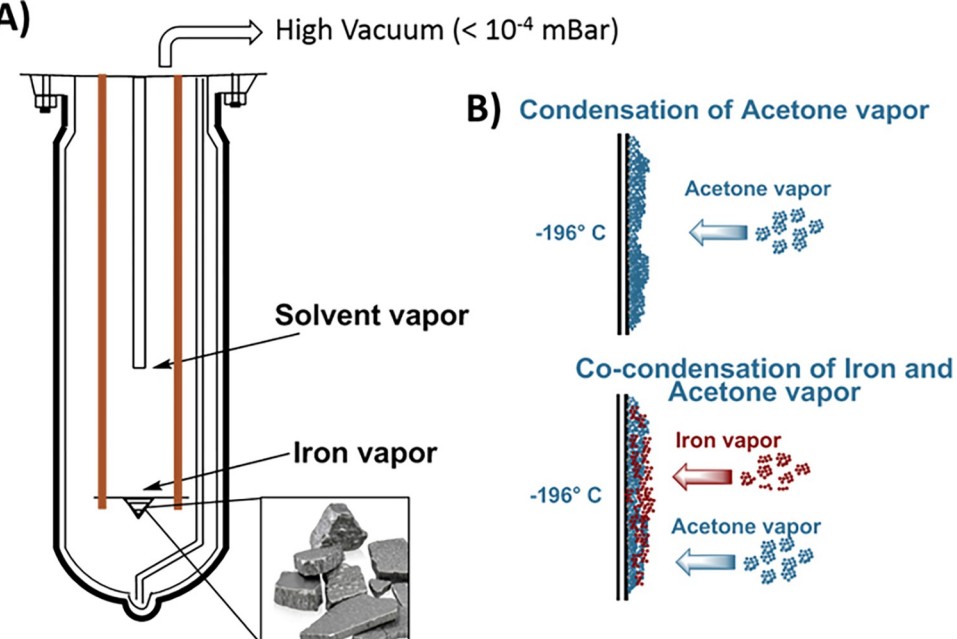

**Fig 1.** Graphical representation of a metal vapor synthesis reactor (A); Main reactions occurring onto the cold (-196˚C) reactor walls during the MVS process (B). Once the evaporation is stopped, the liquid nitrogen bath is removed, the matrix melts and the acetone solution of Fe NPs is collected by siphoning. This solution is thereafter treated with oxygen and glucose, as described in the text.

melted brown SMA solution was siphoned out of the vessel and stored at -40°C. A total of 50 mL of the acetone SMA solution was added to a solution of D-glucose in water (5g in 15 mL) at 0° C under gentle stirring, and the resulting mixture was left for one day at room temperature open to air. In these conditions, a brown precipitate formed, which was collected on a Buchner funnel and washed repeatedly with cold water and acetone. The iron content was measured by ICP-OES, after heating a portion of the solid with aqua regia in a porcelain crucible and dissolving the residue in HCl (dil). The Fe content resulted in a 2.28 percentage weight.

Transmission electron microscopy (TEM) and energy-filtered transmission electron microscopy (EFTEM) analyses were performed on a ZEISS LIBRA200FE microscope equipped with a 200 kV FEG source. The specimens were dispersed in water and sonicated, then each suspension was dropped onto a holey carbon-coated copper grid (300 mesh), and the solvent thus evaporated.

DLS and Z-potential analyses were performed with a 90 PLUS Particle Size Analyzer (Brookhaven Instruments Corporation). A total of 10 mg of Glc-SPIONs were dispersed in 6 mL of deionized water or a solution 0.1 M of $NaHCO_3$ (pH = 8.3) or 1 mM HCl (pH = 3), respectively, and passed through a cellulose acetate filter (0.45 μm).

The stability of the Glc-SPIONs was evaluated. The colloid was stable at room temperature under neutral conditions for more than one week.

## Relaxivity and imaging

Longitudinal relaxivity ($R_1$) was measured at 14.1 T on an Agilent Inova 600. Five solutions with variable nanoparticle concentrations ([Fe] between 1 and 10 mM) were prepared by dissolving them in 0.3M $NaHCO_3$ in $D_2O$ and measuring the HDO-$T_1$ by standard inversion recovery. The relaxation rates for each solution were corrected by subtracting the relaxation rate of the blank.

MR images were taken at 1.5 T on a Siemens Magnetom Avanto scanner.

A T1 sequence was applied: TE = 13 ms TR = 739 ms. A variable number of Eppendorf vials filled with the Glc-SPION solutions at increasing concentrations or with the cell suspensions to be analysed were placed in a rack and scanned.

For the imaging of the Glc-SPION solutions, we used the following concentrations: 1,2,4,6,8,10 mg/mL.

For cell suspension, we used BCPAP cells, which were treated for 6h with different concentrations of Glc-SPION (0.25, 0.50, 1.00 mg/mL). After treatment, the cells were centrifuged and washed to remove the growth medium containing the Glc-SPIONs and resuspended in fresh (Glc-SPION-free) growth medium.

## Cell lines

Pancreatic (PSN-1) carcinoma cell lines as well as human embryonic kidney HEK293 non-tumor cells were obtained from the American Type Culture Collection (ATCC, RocKville, MD). Human thyroid (BCPAP) carcinoma cells were purchased from the DSMZ (Leibnitz Institute DSMZ-German Collection of Microorganisms and Cell Cultures).

Cells were routinely cultured in the following media containing 10% fetal heat-inactivated calf serum (FCS; Euroclone, Milan, Italy), antibiotics (50 units/mL penicillin and 50 mg/mL streptomycin), and 2 mM L-glutamine: i) RPMI-1640, PSN-1 and BCPAP cells; ii) DMEM for HEK293 cells. All cultures were kept at 37°C in a humidified atmosphere with 5% $CO_2$. Cell transfer and preparation of single-cell suspensions were performed by mild enzymatic dissociation using a 0.05% trypsin and 0.02% EDTA solution in PBS (Euroclone, Milan, Italy). Glc-

SPIONs were dissolved just before the experiment in a physiological solution with 0.3M $NaHCO_3$ (pH 7.4) added.

## Cytotoxicity assays

The growth inhibitory effect on cell lines was evaluated by the MTT, as previously described. Cancer cells were seeded in 96-well microplates in growth medium (100 μL, $5x10^3$ cells/well), and then incubated in a 5% carbon dioxide atmosphere at 37˚C. After 24 h, the medium was replaced with a fresh one containing different concentrations of Glc-SPIONs (range 0.01–0.1 mg/mL). Triplicate cultures were established for each treatment. After 1, 3, 6, 12, and 24 h, 10 μL of a 5 mg/mL MTT saline solution were added to each well, and microplates were incubated for an additional five hours. Subsequently, 100 μL of a sodium dodecyl sulfate (SDS) solution in 0.01 M HCl were added to each well. After an overnight incubation, the inhibition of cell growth induced by the tested compound was evaluated by measuring the absorbance at 570 nm, using a BioRad 680 microplate reader (BioRad Laboratories S.r.l.; Segrate, Italy). The average absorbance for each Glc-SPION dose was expressed as a percentage of the control and plotted versus drug concentration. $IC_{50}$ values were obtained from the dose-response curves by means of the 4-PL model ($p < 0.05$). $IC_{50}$ values are the drug concentrations that reduce the mean absorbance at 570 nm to 50% of those of the untreated control wells.

## Evaluation of GLUT1 expression in tested cancer cells

Expression levels in cancer cells were evaluated by means of the GLUT1 Colorimetric Cell-Based ELISA kit (Boster Biological Technology, Pleasanton CA, USA). Approximately $2x10^4$ cells were seeded in 96-well microplates, and GLUT1 expression was detected at 450 nm, using a BioRad 680 microplate reader (BioRad Laboratories S.r.L.), following the manufacturer's instructions.

## Evaluation of Fe uptake in vitro

The cellular uptake of Glc-SPION was evaluated using concentrations and time exposures that did not affect cell viability. Cancer cells ($2.5x10^6$) were seeded in 75 $cm^2$ flasks in growth medium (20 mL). After 24h the medium had been replaced, and the cells were incubated for 30 min together with 1, 3, 6, and 12h with 0.1 mg/mL of the tested Glc-SPIONs. Cells were treated for 30 min, 1, 3, 6, and 12 h with 0.1 mg/mL of Glc-SPIONs, and the Fe content was detected by GF-AAS. Cell monolayers were washed twice with cold PBS (2 mL) and harvested. Samples were subjected to three freeze/thaw cycles at -80˚C and then vigorously vortexed. Aliquots were removed for the determination of protein content by the Bradford protein assay (BioRad). The samples were treated with 1 mL of highly pure nitric acid (Fe: <0.01 mgkg-1, TraceSELECT Ultra, Sigma Chemical Co.), and transferred into a Teflon microwave vessel. Samples were then submitted to a standard procedure using a speed wave MWS-3 Berghof instrument (Eningen, Germany). After cooling, each mineralized sample was analyzed for iron using a Varian AA Duo graphite furnace atomic absorption spectrometer (GF-AAS, Varian, Palo Alto, CA, USA) at 248.3 nm. The calibration curve was obtained using known concentrations of standard solutions purchased from Sigma Chemical Co. The results were expressed as ppb of Fe per mg of protein [28].

In competition uptake experiments, before exposure to the tested Glc-SPIONs, cells were pre-incubated for 1 h with GLUT1 inhibitors, namely anti-GLUT1 polyclonal antibody (Abcam, Cambridge, UK), WZB117 (3-fluoro-1,2-phenylene bis 3-hydroxybenzoate), 3-Hydroxy-benzoic acid 1,1′-(3-fluoro-1,2-phenylene ester, Sigma Aldrich), Fasentin (N-[4-Chloro-3-(trifluoromethyl)phenyl]-3-oxobutanamide, Sigma Aldrich), BAY-876 (N4-[1-

[(4-Cyanophenyl)methyl]-5-methyl-3-(trifluoromethyl)-1H-pyrazol-4-yl]-7-fluoro-2,4-quino-linedicarboxamide, Sigma Aldrich), STF-31 (Aurogene, Rome, Italy).

## Experiments with animals

All experiments were performed according to Italian law (D.L.vo 26/2014), which regulates the use of experimental animals in Italy. The research project was approved by the Italian Health Department in accordance with art. 20 of the above-mentioned D.L.vo. The mice were purchased from Charles River, Italy, housed in steel cages under controlled environmental conditions (constant temperature, humidity, and 12 h dark/light cycle), and fed with standard commercial feed and tap water ad libitum.

Preliminary biodistribution studies were performed in mice inoculated with Lewis Lung Carcinoma (LLC), as an example of the syngenetic murine model. The LLC cell line was purchased from ECACC (United Kingdom). The cells were maintained in DMEM (Euroclone, Pero, Italy) supplemented with 10% heat-inactivated fetal bovine serum (Euroclone, Pero, Italy), 10 mM L-glutamine, 100 U/mL penicillin, and 100 μg/mL streptomycin in a 5% $CO_2$ air incubator at 37˚C. The LLC was implanted intramuscularly (i.m.) as a $2 \times 10^6$ cell inoculum into the right hind leg of 8-week-old male and female C57BL mice ($24 \pm 3$ g body weight).

The well-being of the animals was monitored every day assessing physiological parameters, such as body weight, abnormal posture, and respiratory signs. Nine days after tumor implantation (tumor volume of about 100 mm$^3$: the long axis (L) and the short axis (S) were measured with caliper, and the tumor volume (V) was calculated using the following equation: V = SxSxL/2), mice were randomly allocated to four groups (five animals per group), anesthetized (with Zoletil$^{®}$ 40 mg/kg and Rompun$^{®}$ 10 mg/kg i.p.) and treated with Glc-SPION (1 mg/kg) administered i.v.. Control mice received the vehicle solution (saline). After 1, 3, 6, and 24 h animals were sacrificed (by $CO_2$ overdose) and tumor, kidney, intestine, lung, and liver were excised, while urine and blood were collected and stored at −20˚C. Tissues were subsequently mineralized in $HNO_3$. Fe content of each sample was measured by GF-AAS as described above. Results were expressed as % of Fe (% of injected).

## Evaluation of nephrotoxicity

Nephrotoxicity was assessed according to previous experiments by Gandin et al. [34]. To gain more insight into their putative nephrotoxicity potential, mice treated with a single i.v. injection of Glc-SPIONs were evaluated in terms of induction of acute kidney damage. Cisplatin was used as a positive control, as it is well known that cisplatin causes irreversible renal damage.

The potential nephrotoxic effect was evaluated by measuring specific biomarkers in the urine samples. Urine samples were collected from the treated animals after 12, 24, and 72 h, and urinary total protein (uTP) and N-acetyl-β-D-glucosaminidase (NAG) were evaluated as signs of nephrotoxicity.

Mice treated with a single i.v. injection of Glc-SPION (1 mg/kg) or the vehicle solution (0.2 mL saline solution, control) were placed in metabolic cages, and urine samples were collected after 12, 24, and 72 h. Cisplatin (3 mg/kg) i.v. was used under the same experimental conditions as the positive control. Urine samples were centrifuged (150$g$ for 10 minutes at room temperature) to discard debris, and aliquoted to measure creatinine, uTP, and NAG. Urine creatinine assays were performed using the creatinine assay kit from Sigma Chemical Co. (St. Louis, MO). uTPs were measured using the BioRad Total Protein Test (Hercules, CA). Urinary NAG was measured spectrophotometrically with the NAG kit (Roche Diagnostics,

Basel, Switzerland) according to the manufacturer's protocols. uTP and NAG were expressed as grams per millimoles of creatinine (g mmol$^{-1}$ creatinine).

### Statistical analysis

All values were the means ± SD of no less than three measurements from three different cell cultures. Multiple comparisons were performed by ANOVA, followed by the Tukey–Kramer multiple comparison test ([**]$P < 0.01$; [*]$P < 0.05$), using GraphPad.

## Results

### Structural characterization of Glc-SPIONs

High resolution TEM analysis (Fig 2A) revealed the presence of small roundish crystalline iron oxide nanoparticles in the organic amorphous phase of glucose, with no evidence of the generation of metal aggregates in the scanned area. The particles were distributed in a narrow range (1.5 nm—3.5 nm) with a mean diameter of 2.7 nm.

The presence of iron in the material was also investigated by the energy-filtered TEM (EF-TEM) technique, where the image is formed by detecting any electrons that lost energy during interaction with the sample (Fig 2A). The iron map (Fig 2B) was obtained by detecting electrons with an energy loss of 713 eV, (typical of iron L$_{2,3}$ peak). It shows that the iron signal coincides with the NPs, as seen in the corresponding reference image collected at low magnification (Fig 2C). Unfortunately, the intensity of the iron signals in the EF-TEM iron map was too low to reliably identify the NPs within the composite, nevertheless the analysis confirmed the presence of a highly homogeneous iron phase dispersion within the glucose matrix.

The hydrodynamic mean diameter of the Glc-SPIONs, measured by DLS, was 15.5 nm, thus confirming the presence of the organic glucose shell around the iron oxide core. By combining TEM and DLS data, the thickness of the organic shell in solution was estimated as being around 6.4 nm. The surface charge of the Glc-SPIONs measured by z-potential was -26.44 mV, in neutral conditions, confirming the high degree of stability of the nanoparticle dispersion (Fig 2F). A zeta potential of– 39.6 and– 0.3 mV was recorded at pH 8.3 and 3, respectively, proving the presence of the negative charge on the surface of the nanoparticles.

Fig 2D and 2E show the histograms of the particle size distribution and the hydrodynamic diameter distributions measured by DLS.

### Preliminary magnetic resonance imaging experiments

With a value of R1 = 0.084 s-1 mM-1, we hypothesized that our Glc-SPIONs would exert a modest positive contrast in T1 MRI images. In order to verify this, we first placed six Eppendorf vials into an MRI scanner. The vials contained increasing concentrations of Glc-SPIONs (from 0 to 10 mg/mL). The vials were placed in a plastic rack and subjected to a T1 sequence. From 4 mg/mL onwards, a constant level of positive contrast was clearly shown in a T1-weighted sequence (Fig 3).

Once the contrast in the simple buffer solution had been demonstrated, a similar test was conducted on a set of tumor cell dispersions, previously treated with increasing concentrations of Glc-SPIONs. Cells were treated for 6h, centrifuged and washed three times and resuspended in fresh medium. Four samples, including a control experiment with non-treated cells, were subjected to the same T1-sequence.

As shown in Fig 3, the control sample consisting of non-treated cells appears in dark grey, whereas the three vials containing cells treated with Glc-SPIONs are uniformly bright, revealing a positive contrast: in these conditions the cells had absorbed Glc-SPIONs to saturation.

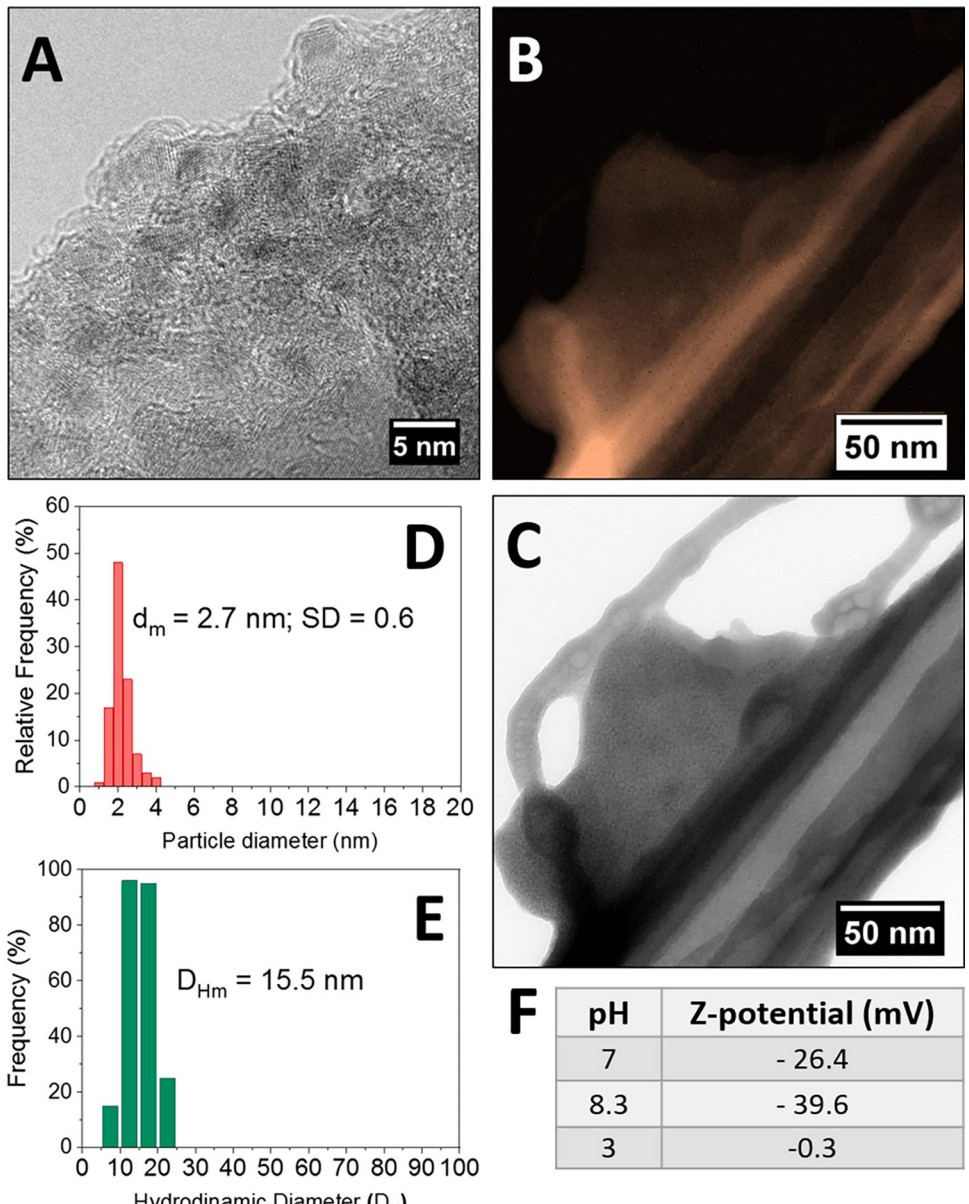

**Fig 2.** HR-TEM image of Glc-SPIONs (A); false-color EF-TEM images filtered at 713 eV providing Fe maps (B); reference TEM image (C); histogram of particle size distribution (D) and histogram of hydrodynamic diameter distributions (measured by DLS) (E); value of zeta potential of– 39.6 and– 0.3 mV at pH 8.3 and 3 (F).

The contrast effect exerted in cell suspension is much more pronounced than in a simple solution. In fact, even at the lowest concentration (0.25 mg/mL), a bright image was visible, comparable to 8 mg/mL in water (Fig 3). This could be the result of the efficient Fe uptake by BCPAP cells, possibly associated with an increased relaxivity of Glc-SPIONs in the more viscous cellular solution.

## Cytotoxicity in human cancer cells

As reported in Fig 4, the cell viability assay indicated a very low cytotoxicity in the range of concentrations in all the cancer cell lines tested.

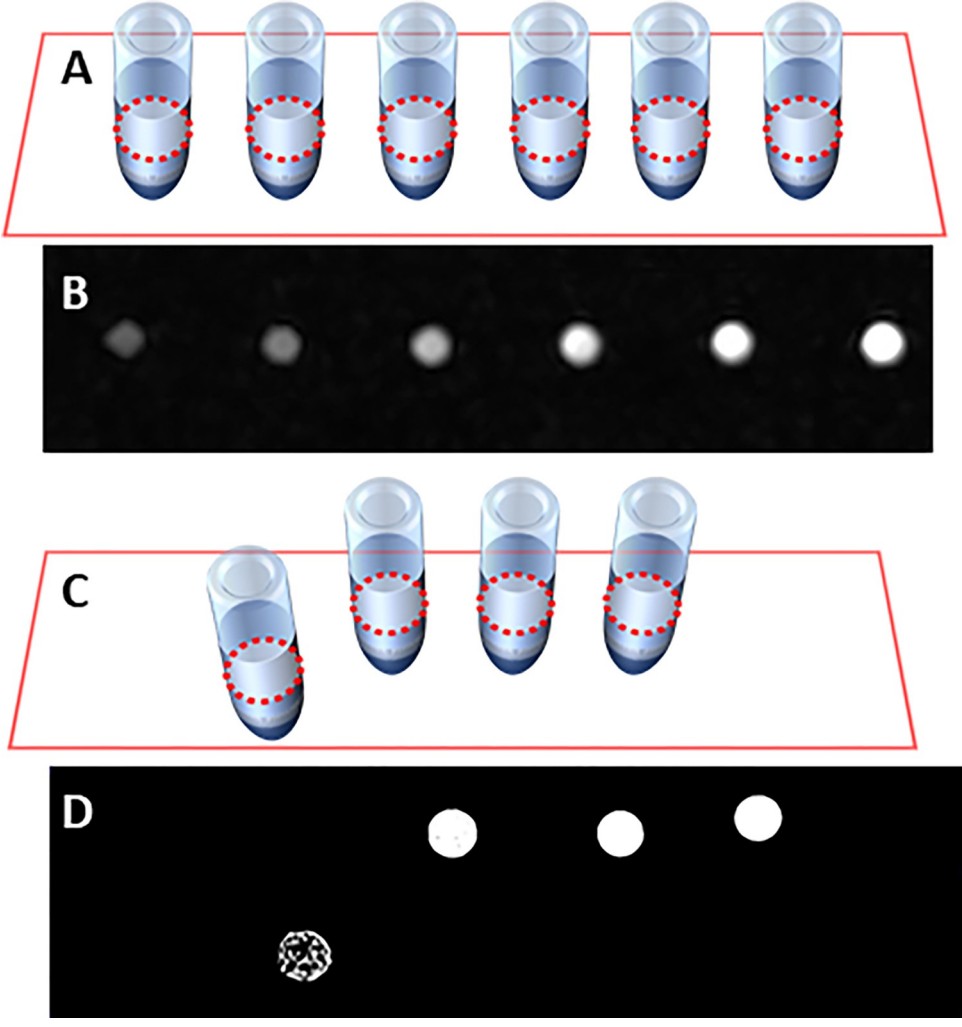

**Fig 3.** A) Schematic representation of the arrangement of the set of 6 Eppendorf vials filled with (from left to right): 0, 1, 2, 4, 8, 10 mg/mL Glc-SPIONs in 0.1 M NaHCO3; B) T1-weighted sagittal image of the rack of vials; C) Schematic arrangement of the skewed set of 4 Eppendorf vials, containing a suspension of cells treated for 6 hours with (from left to right): 0, 0.25, 0.50, 1 mg/mL Glc-SPIONS; the sample out of line represents the blank; D) T1-weighted sagittal image of the set of 4 Eppendorf vials.

## GLUT1 expression in cancer cells

Fig 5 highlights that the cancer cells tested expressed GLUT1, and that the highest GLUT1 expression level was found in pancreatic PSN-1 cancer cells, in which GLUT1 detected levels where 3.5 times higher than in the BCPAP cells. In contrast, non-cancer human HEK293 cells presented significantly lower GLUT1 levels compared to all the pancreatic PSN-1 cancer cells, and slightly lower levels compared to human thyroid BCPAP cancer cells.

## Uptake in cancer cells

Fig 5 shows that Glc-SPIONs are internalized in all the cancer cells in a time-dependent manner. In particular, PSN-1 cancer cells were the most effective at internalizing Glc-SPIONs. Although significantly higher than the control cells, a lower Fe content was detected in human BCPAP thyroid cancer cells treated with Glc-SPIONS. Notably, Glc-SPION uptake in the two

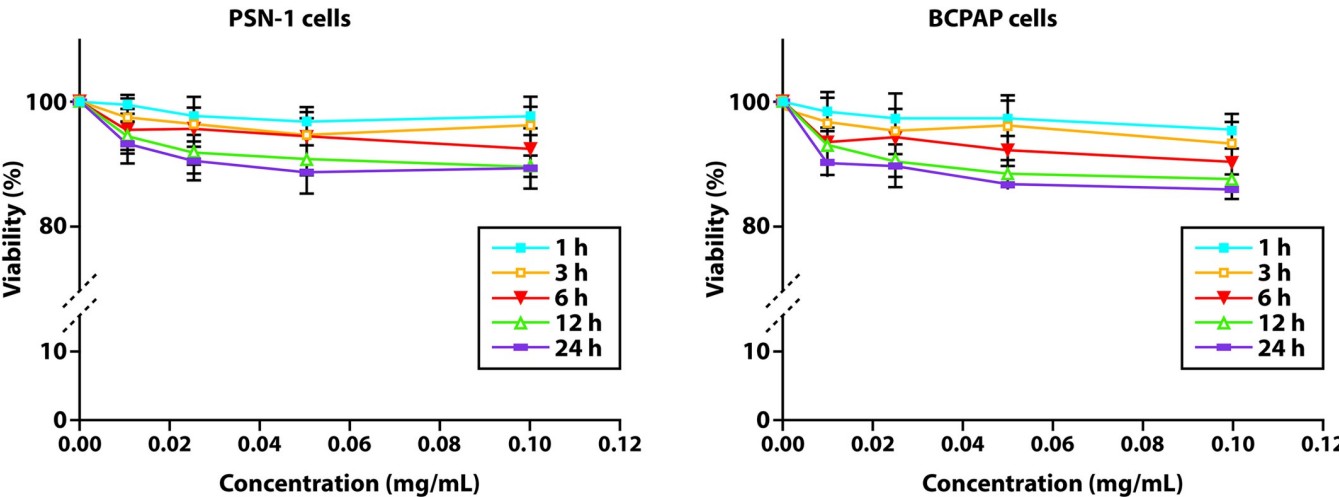

**Fig 4. Cytotoxicity in human pancreatic and thyroid cancer cells.** Cells ($5 \cdot 10^3 \cdot mL^{-1}$) were treated for 1, 3, 6, 12, and 24 h with increasing concentrations of Glc-SPIONs. Cytotoxicity was assessed by the MTT test. $p < 0.05$.

different cancer cells correlated well with the detected GLUT1 levels, thus suggesting the involvement of GLUT1 in the cellular internalization of Glc-SPIONS.

To confirm the involvement of GLUT1 in Glc-SPIONS internalization, cellular uptake experiments were also conducted by pre-treating cancer cells for 1 h with specific GLUT1 inhibitors, namely a polyclonal anti-GLUT1, WZB117, Fasentin, BAY-876, and STF-31. These results are reported in Fig 5.

Although to differing extents, all the inhibitors reduced the cancer cell uptake of Glc-SPIONS, mostly at six hours ($p < 001$). This effect was much more evident in human lung PSN-1 cells whereas it was slightly evident, but significant, in human thyroid BCPAP cancer cells. These results strongly suggest the involvement of GLUT1 in the internalization mechanism of Glc-SPIONS in cancer.

## Preliminary biodistribution studies

Fig 6 panel A shows the optimal biodistribution of Glc-SPIONS. After 1 h, a higher content of Fe was detected in the blood, followed by the urine and the tumor. In contrast, very low levels of Fe were detected in the lung and intestine. After 3 h, iron was found at high levels in the urine, blood, and tumor. The levels of Fe in the tumor mass were similar after 6 h, whereas the iron content significantly dropped in the blood samples, but increased in the urine. These results clearly indicate a great bioavailability to the malignant tissue by the i.v. administration of Glc-SPIONS. Interestingly, a substantial number of Glc-SPIONS were excreted in the urine 6 h after injection, thus supporting the hypothesis that Glc-SPIONS can be efficiently eliminated by the kidney.

## Evaluation of nephrotoxicity

As expected, cisplatin induced a significant increase in uTP excretion (Fig 6, panel B) and NAG (Fig 6, panel C). In contrast, treatment with Glc-SPIONS led to a 24 h excretion of uTP, which was roughly three times lower than the one recorded with cisplatin. In the following 72 h, the levels of uTP excreted after Glc-SPION administration were about four times lower compared to those of the metallodrug.

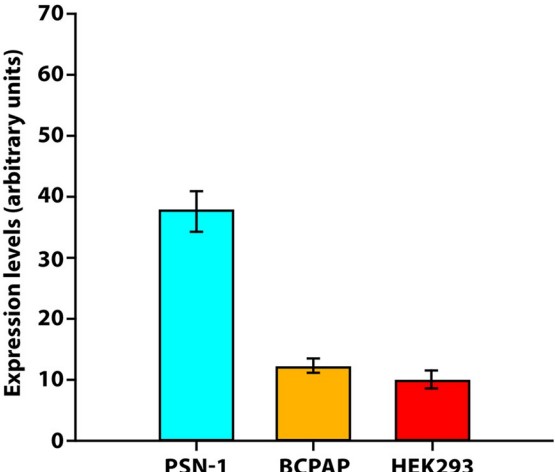

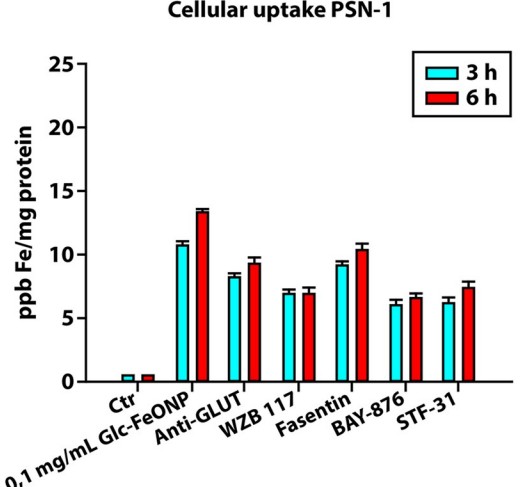

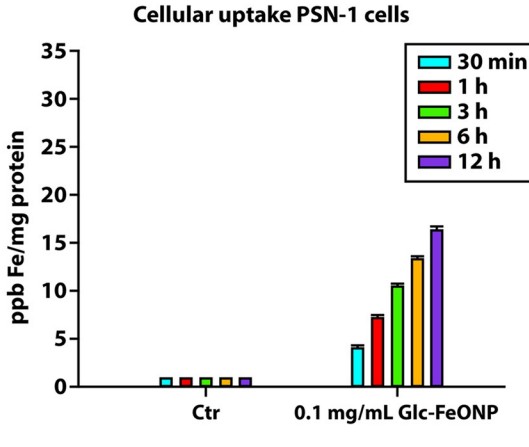

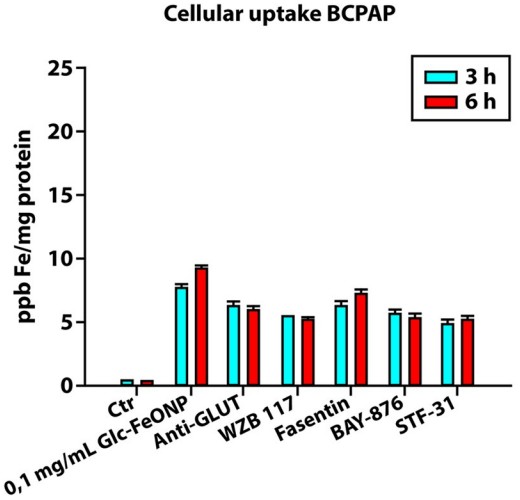

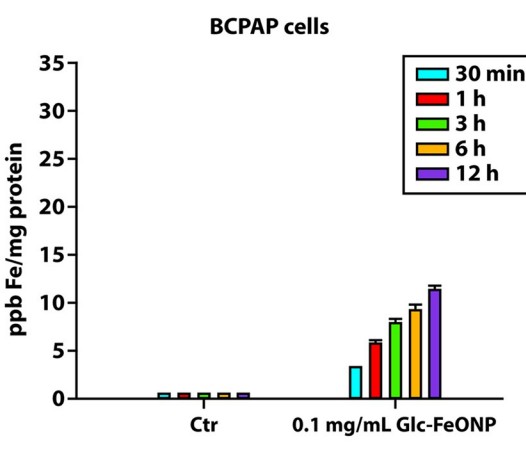

**Fig 5. Internalization studies of Glc-SPIONs.** GLUT1 levels estimated by ELISA in PSN-1, BCPAP and HEK293 cells (at the top). Uptake studies: PSN-1 and BCPAP cells were incubated with 0.1 mg/mL of Glc-SPIONs for 30 min, 1, 3, 6, and 12 h. The Fe cellular content was estimated by GF-AAS analysis. Uptake studies after treating the cells with GLUT1 inhibitors: PSN1 and BCPAP cells were pre-incubated for 1 h with anti-GLUT1 polyclonal antibody, WZB117, Fasentin, BAY-876, and STF-31. Subsequently, cells were treated for 3 or 6 h with 0.1 mg/mL of Glc-SPIONs.

NAG activity detected after injection with Glc-SPIONs was approximately two times lower than that detected after the injection of cisplatin (Fig 6, panel C). These results clearly suggest the lower nephrotoxic potential of Glc-SPIONs compared to the clinically approved metallo-drug, cisplatin.

## Discussion

Delivering SPIONs to tumor cells by i.v. administration could represent "the magic bullet" for detecting and treating cancer.

There are some examples in the literature of MNs that are decorated on their surface with specific elements and ligands in order to be recognized by cancer cell receptors, thus allowing

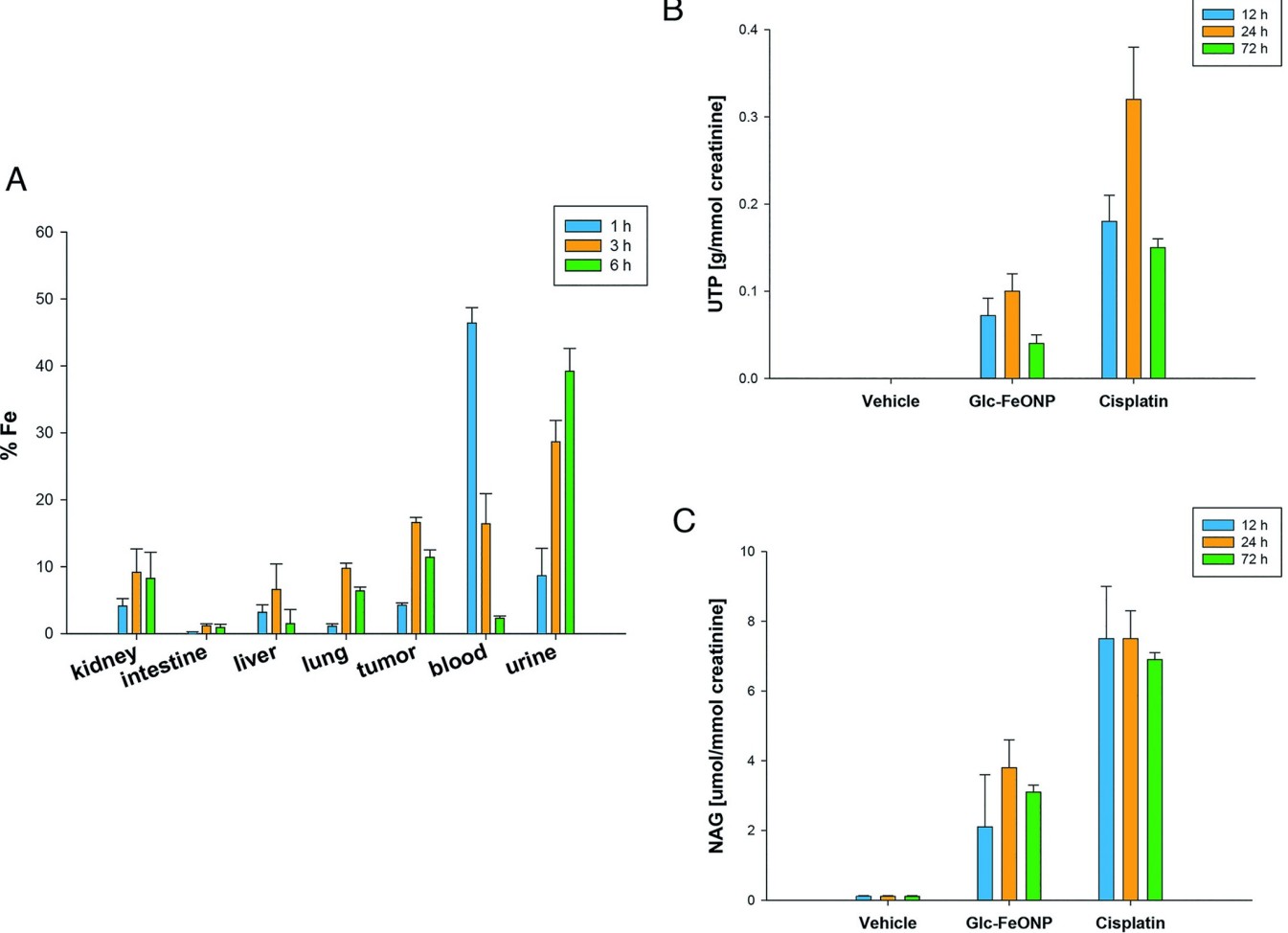

**Fig 6. Preliminary *in vivo* studies. A.** Biodistribution studies. Total iron levels determined in the organs or urine of mice treated with Glc-SPIONs after a single i.v. dose (1 mg kg$^{-1}$). **B** and **C.** Nephrotoxicity studies. Mice were treated with a single i.v. injection of Glc-SPIONs (1 mg kg$^{-1}$), CDDP(3 mg kg$^{-1}$) or the vehicle solution (saline solution, control) and were then placed in metabolic cages. Urine samples were collected after 12, 24, and 72 h, centrifuged and aliquoted to measure creatinine, uTP (B), and NAG (C). The error bars indicate the SDs of not less than three measurements (p0.01).

selective targeting for some specific cancers [19,27,28]. We focused on glucose coated MNs since this simple molecule could represent a "universal" ligand for cancer [29]. We also discovered that glucose provides a stable coating for small iron oxide superparamagnetic nanoparticles, obtained by MVS.

Glucose is present at an early stage of nanoparticle preparation, and participates intimately in their structure. Any different stabilizing agent would result in different particles. The nature of the interaction between glucose and the iron oxide core merits further investigation, however it has been shown to result in stable and clear suspensions of Glc-SPIONs in buffered solutions. Dextran is a glucose polymer and is an excellent and widely used coating for iron oxide nanoparticles, because polyol groups are good stabilizers for the core. However, owing to its polymeric nature, it cannot be recognized by glucose receptors [30].

The properties of our Glc-SPIONs enable the selective targeting of cancer cells and also include low toxicity, as well as good magnetic properties. In addition, for practical usage, the MVS route to Glc-SPIONs is relatively easy and produces homogeneous nanoparticles with a total absence of contaminants, because the only reactants in play are heat (which vaporizes Fe), air (which provides oxygen), and glucose. Acetone is used at an early stage of preparation, but is completely removed thereafter.

In a previous paper [31], we addressed the problem of glucose being responsible for cell uptake, by comparing Glc-SPIONs with polyvinylpyrrolidone-coated SPIONs (PVP-SPIONs) as a control group. In fact, that experiment demonstrated that PVP is a good stabilizer, however it leads to SPIONs which cannot elicit any active uptake into cells and are endowed with different physico-chemical characteristics. We thus felt that PVP-SPIONs are not suitable for a control experiment and preferred to use different GLUT1-inhibitors. We also used different cell lines that show notably different expressions of GLUT1. In fact, we investigated two neoplastic lines, specifically pancreatic cancer (PSN-1), human thyroid cancer (BCPAP), and the non-tumor human embryonic kidney HEK293 cell line.

The uptake of Glc-SPIONs in the various cancer cell lines correlated well with the GLUT1 levels detected, thus suggesting the involvement of GLUT1 in the cellular internalization of Glc-SPIONs. The involvement of GLUT1 in Glc-SPION internalization was corroborated through cellular uptake tests by pre-treating cancer cells for 1 h with different specific GLUT1 inhibitors, which acted in different ways. In fact, we used polyclonal anti-GLUT1, WZB117, Fasentin, BAY-876 and STF-31, and all the results were in agreement showing that all these substances could produce a significant inhibition of Glc-SPION uptake.

To differing extents, all the inhibitors reduced the cancer cell uptake of Glc-SPIONs, and the degree of inhibition was proportional to the expression of GLUT1, which also in this case, was more evident in human PSN-1 cancer cells, and less evident in human papillary thyroid BCPAP cancer cells.

At this point we can state that the transport inside the cells is mediated by GLUT1, but we cannot make any speculations regarding the intimate endocytosis mechanism by which our MNs enter the cells.

To the best of our knowledge, this is thus the first paper in which the inhibition of Glc-SPIONs has been robustly demonstrated, with specific inhibitors displaying different mechanisms of action. All these data offer the clearest evidence so far that glucose-coated MNs can be electively internalized in cancer cells via the GLUT1 transporter, although we cannot rule out that other glucose transporters may also be active.

Although we were unable to take MRI scans of the animals, we demonstrated that, even at the lowest dose, the active internalization on malignant cells is such that a strong positive contrast is obtained, which is the necessary premise for the use of Glc-SPIONs as a diagnostic tool.

We tried using Glc-SPIONs for magnetic induction hyperthermia, however, unfortunately, the experiments were not successful due to the very small size of the SPIONs, which led to less than optimal, magnetic properties. We are currently working on three different strategies: obtaining larger particles; co-evaporating two different metals to prepare bimetallic nanoparticles; and modulating the oxidation conditions to change the iron oxide composition. These are all aimed at improving the magnetic properties.

Larger particles can be both a goal and a problem at the same time. On the one hand they present improved magnetism, which means a higher MRI contrast and possibly better performance in hyperthermia. On the other, they may be less able to penetrate the cell membrane. The aim, therefore, is to find the right balance.

Our data regarding biodistribution and toxicity appear promising. In fact, after Glc-SPION i.v. administration, the level of Fe in blood rapidly decreased over time, while the Fe content increased in parallel in the urine. Fe concentrations appeared in the liver, lung, and, of course, the kidney, although a rapid decrease in Fe content was observed, becoming almost negligible at 6 hours in the lung and intestine.

In contrast, Fe concentrated well in the tumor, which maintained a high concentration even after six hours. In fact, six hours after administration, apart from the urine, and to a lesser extent the kidney, the largest amount of Fe was in the tumor. As already stated, we do not currently know how endocytosis works in detail. However, considering the role of GLUT1, the trafficking from the membrane to the inner part of the cell may perhaps be mediated by chlatrin and this could be a further issue of investigation.

Our results indicate the great bioavailability of Glc-SPIONs to the malignant tissue by i.v. administration. The fact that a substantial proportion of Glc-SPIONs were excreted via urine 6 h after injection supports the hypothesis that Glc-SPIONs can be efficiently eliminated by the kidney. In addition, we investigated their putative nephrotoxicity potential by treating mice with a single i.v. injection of Glc-SPIONs in order to evaluate acute kidney damage. Our other important goal, in fact, was to limit the nephrotoxic effect.

Our results show that all the parameters of renal damage were much lower than those detected after compared to that shown by the clinically approved metal-based drug.

## Conclusions

The properties of our SPIONs enable the selective targeting of cancer cells. They have low levels of toxicity, as well as good magnetic properties We are currently working on overcoming some limitations of this preliminary work. However, we strongly believe that our results constitute preliminary evidence that MVS provides a powerful route to prepare small SPIONs, and that they can be coated and stabilized by a glucose shell. This would enable one of the most universal metabolic anomalies of cancer to be exploited: the Warburg effect.

Our Glc-SPIONs were efficiently internalized in malignant cells and tissues, ensuring at the same time optimal biodistribution after systemic administration, without impairing cells and animal viability. Promising MRI evidence indicates that Glc-SPIONs can be used as a positive MRI contrast, while some more optimization is required to achieve magnetic hyperthermia. To this end, one of the advantages of MVS consists in the possibility to change the ratio between the glucose shell and the metal core and even the composition of the core itself. Finally, in our view, our SPIONs appear to have many of the prerequisites of an ideal MN in the oncological field.

## Author Contributions

**Conceptualization:** Daniele Barbaro, Andrea Ciaramella.

**Investigation:** Valentina Gandin, Cristina Marzano, Michele Malventi, Claudio Evangelisti.

**Methodology:** Lorenzo Di Bari.

**Writing – original draft:** Daniele Barbaro, Lorenzo Di Bari, Valentina Gandin.

**Writing – review & editing:** Daniele Barbaro, Lorenzo Di Bari, Valentina Gandin.

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
