## [Decision Letter · Decision Letter 0]

11 Oct 2021

PONE-D-21-21393Glucose-coated superparamagnetic iron oxide nanoparticles prepared by metal vapor synthesis can target GLUT1 overexpressing tumors: in vitro tests and in vivo preliminary assessmentPLOS ONE

Dear Dr. Barbaro,

Thank you for submitting your manuscript to PLOS ONE. After careful consideration, we feel that it has merit but does not fully meet PLOS ONE’s publication criteria as it currently stands. Therefore, we invite you to submit a revised version of the manuscript that addresses the points raised during the review process.

We look forward to receiving your revised manuscript.

Kind regards,

Irina V. Balalaeva, PhD

Academic Editor

PLOS ONE

2. To comply with PLOS ONE submissions requirements, in your Methods section, please provide additional information regarding the experiments involving animals and ensure you have included details on (1) methods of sacrifice, (2) methods of anesthesia and/or analgesia,  (3) efforts to alleviate suffering  4) maximum tumor size allowed and 5)how you assessed animal well-being and how often.

3.Thank you for stating the following financial disclosure:

“No”

“No”

Additional Editor Comments (if provided):

Reviewers' comments:

Reviewer's Responses to Questions

**Comments to the Author**

1. Is the manuscript technically sound, and do the data support the conclusions?

Reviewer #1: Yes

Reviewer #2: Yes

2. Has the statistical analysis been performed appropriately and rigorously? 

Reviewer #1: I Don't Know

Reviewer #2: Yes

3. Have the authors made all data underlying the findings in their manuscript fully available?

Reviewer #1: Yes

Reviewer #2: Yes

4. Is the manuscript presented in an intelligible fashion and written in standard English?

Reviewer #1: No

Reviewer #2: Yes

5. Review Comments to the Author

Reviewer #1: The manuscript needs English polishing. Also needs better discussion. I suggest major revision

1-Add comma after paper: In this paper we show the

2-Introduction needs revision. There are some self-citation in the introduction and they can be managed better to be cited. Other studies should be added, e.g., Nano Today 2021, 38, 101119

3-Please add ref for this sentence: Often, polymeric coatings are used, 53 which create a barrier between the MN core and the biological fluid, resulting in extracellular fluid

contrast agents, an application in which gadolinium (Gd) complexes are most widely used. Recommended ref: Molecules 2020, 25 (18), 4053; Advanced Functional Materials 2020, Vol 30, 22 1910021.

4-Figure 2 and 3 can be merged. Also need scale bar.

5-The graphs were not illustrated well. Also would be better to be colorful + having pattern. Therefore, it will be suitable for both B/W and colorful printing.

6- Text correction: There are some typos and misspelling that should be corrected

7-A schematic illustration is needed on the reaction/preparation of the NPs to help the readers to understand the process.

8-ToC graphical abstract could be added to improve the manuscript.

9-the manuscript lacks discussion. In most cases, only the results were elaborated without to be compared with other results. For instance, this paragraph need discussion: "In contrast, Fe concentrated well in the tumor, which maintained a high concentration also after 6 ...." recommended literature: Nano Today 2021, 40, 101279.

10-References: Many of the used references are up to date which is a positive point but please remove/replace the outdated ones.

13) Ma, H. L., Qi, X. R., Maitani, Y. & Nagai, T. Preparation and characterization of superparamagnetic iron oxide nanoparticles stabilized by alginate. Int. J. Pharm. 333, 177–186 (2007).

15) Xie, J. et al. Human serum albumin coated iron oxide nanoparticles for efficient cell labeling. Chem. Commun. 46, 433–435 (2010).

20) Kayal, S. & Ramanujan, R. V. Doxorubicin loaded PVA coated iron oxide nanoparticles for targeted drug delivery. Mater. Sci. Eng. C 30, 484–490 (2010)

Reviewer #2: In this manuscript, the authors fabricated glucose coated superparamagnetic iron oxide nanoparticle which can target tumor cells based on the affinity of tumor cells for glucose. Following points should be clearly addressed before publication.

(1) The Zeta potentials of the glucose coated superparamagnetic iron oxide nanoparticles in different media should be summarized in a table.

(2) The long time colloidal stability of the glucose coated superparamagnetic iron oxide nanoparticles should be studied.

6. PLOS authors have the option to publish the peer review history of their article (what does this mean?). If published, this will include your full peer review and any attached files.

Reviewer #1: No

Reviewer #2: No

---

## [Author Response · Author response to Decision Letter 0]

4 May 2022

General requests

In the Methods section we have provided additional information regarding the well being animals. 

REVIEWER 1

1)We have re-edited manuscript by an experienced mother tongue ( “E4ACEnglish for Academics” sas di Adrian John Wallwork Via Carducci 9, 56127 Pisa Tel. 340 7888 304 (no fax) adrian.wallwork@gmail.com p.iva / c.f. 01923950503 REA PI – 166038)

2)We have re-edited the introduction: We think to realize that Reviewer asks for a starting with a more generical setting regarding the nano-therapy for cancer and for this reason we have modified the beginning of the introduction and quoted the paper on auto-assembled peptides as suggested. Moreover we have edited and streamlined the whole introduction deleting some parts.

3) We have added the references asked by the Reviewer and we have added some new references.

4) We have merged the figures as suggested

5) We have realized that the graphs hadn’t a not so good quality so we have completely re-done color graphs

6) We have corrected the text

7) We have added an illustration as requested

8)We have added a graphical abstract

9) We have added the reference and implemented the discussion following it

10) we have deleted old references as suggested

In yellow the new we parts and in reds the deleted parts throughout the “ not cleaned version “ of the manuscript

REVIEWER 2 

According to the reviewer comment, the zeta potentials of the glucose coated superparamagnetic iron oxide nanoparticles at different pH values (7, 8.3, 3) have been measured. The results have been summarized in the table reported in Fig. 2 F and included into the manuscript according to that reported below. We have reported that the colloid were stable at room temperature.

In yellow the new we parts and in reds the deleted parts throughout the “ not cleaned version “ of the manuscript

---

## [Decision Letter · Decision Letter 1]

25 May 2022

Glucose-coated superparamagnetic iron oxide nanoparticles prepared by metal vapor synthesis can target GLUT1 overexpressing tumors: in vitro tests and in vivo preliminary assessment

PONE-D-21-21393R1

Dear Dr. Barbaro,

We’re pleased to inform you that your manuscript has been judged scientifically suitable for publication and will be formally accepted for publication once it meets all outstanding technical requirements.

Kind regards,

Irina V. Balalaeva, PhD

Academic Editor

PLOS ONE

Additional Editor Comments (optional):

Reviewers' comments:

Reviewer's Responses to Questions

**Comments to the Author**

1. If the authors have adequately addressed your comments raised in a previous round of review and you feel that this manuscript is now acceptable for publication, you may indicate that here to bypass the “Comments to the Author” section, enter your conflict of interest statement in the “Confidential to Editor” section, and submit your "Accept" recommendation.

Reviewer #2: All comments have been addressed

2. Is the manuscript technically sound, and do the data support the conclusions?

Reviewer #2: Yes

3. Has the statistical analysis been performed appropriately and rigorously? 

Reviewer #2: (No Response)

4. Have the authors made all data underlying the findings in their manuscript fully available?

Reviewer #2: (No Response)

5. Is the manuscript presented in an intelligible fashion and written in standard English?

Reviewer #2: Yes

6. Review Comments to the Author

Reviewer #2: (No Response)

7. PLOS authors have the option to publish the peer review history of their article (what does this mean?). If published, this will include your full peer review and any attached files.

Reviewer #2: **Yes: **Garima Agrawal

---

## [Editor Report · Acceptance letter]

30 May 2022

PONE-D-21-21393R1 

Glucose-coated superparamagnetic iron oxide nanoparticles prepared by metal vapor synthesis can target GLUT1 overexpressing tumors: in vitro tests and in vivo preliminary assessment 

Dear Dr. Barbaro:

I'm pleased to inform you that your manuscript has been deemed suitable for publication in PLOS ONE. Congratulations! Your manuscript is now with our production department. 

Kind regards, 

on behalf of

Dr. Irina V. Balalaeva 

Academic Editor

PLOS ONE